# Chronological Analysis of First-in-Class Drugs Approved from 2011 to 2022: Their Technological Trend and Origin

**DOI:** 10.3390/pharmaceutics15071794

**Published:** 2023-06-22

**Authors:** Ryo Okuyama

**Affiliations:** College of International Management, Ritsumeikan Asia Pacific University, Beppu 874-8577, Japan; ryooku@apu.ac.jp

**Keywords:** first-in-class, modality, target family, origin, small & medium enterprise

## Abstract

The discovery and development of first-in-class (FIC) drugs are becoming increasingly important due to increasing reimbursement pressure and personalized medication. To investigate the technological trends and origin of FIC drugs, the FIC drugs approved in the U.S. from January 2011 to December 2022 were analyzed. The analysis shows that previous major target families, viz. enzymes, G-protein coupled receptors, transporters, and transcription factors, are no longer considered major in recent years. Instead, the shares of secreted proteins/peptides and mRNAs have continuously increased from 2011–2014 to 2019–2022, suggesting that the target family of FIC drugs has shifted to molecules previously considered challenging as drug targets. Small molecules were predominant in 2011–2014, followed by a large increase in antibody medicines in 2015–2018 and further diversification of antibody medicine modalities in 2019–2022. Nucleic acid medicine has also continuously increased its share, suggesting that diversifying modalities supports the creation of FIC drugs toward challenging target molecules. Over half of FIC drugs were created by small and medium enterprises (SMEs), especially young companies established in the 1990s and 2000s. All SMEs that produced more than one FIC drug approved in 2019–2022 have the strong technological capability in a specific modality. Investment in modality technologies and facilitating mechanisms to translate academic modality technologies to start-ups might be important for enhancing FIC drug development.

## 1. Introduction

Newly approved drugs are categorized into two groups, first-in-class (FIC) and follow-on drugs. FIC drugs target molecules that have not been targeted by preceding drugs and provide a novel mechanism of action, adding to new therapeutic options for patients. The entry of follow-on drugs into the FIC drug class became less common after the 2000s, and the motivation for FIC drug development has been increasing [1]. This is partly due to the increasing reimbursement pressure on drugs that lack better profiles than existing drugs [2]. The demand for “me-too” drugs that show minimum differentiation over preceding drugs is reduced, and innovative drugs that provide advantages (better efficacy, better safety, etc.), over existing drugs are desired. Additionally, many one-fits-all-type blockbuster drugs have been developed and have become generic at lower prices, further increasing the demand for developing innovative drugs. Medicines that are more personalized for each patient are currently desired [3,4]. These situations also increase the demand for new mechanisms that meet each patient’s needs. Further, the surge in the desire for FIC drugs is caused by increased orphan drug development for the treatment of rare diseases. A total of 410 drugs approved by the United States Food and Drug Administration (FDA) from 2011 to 2020 contain 184 orphan drugs, and more than half of the FIC drugs approved in the same duration were developed as orphan drugs [5]. To date, more than 7000 rare diseases are reported, most of which do not have effective therapeutic options. Therefore, innovative medicines are desired for treating many rare diseases [6]. Many FIC drugs have been developed as orphan drugs in recent years, and various drug modalities have increased the opportunity for developing new drugs for rare diseases [7,8]. Therefore, the importance of FIC drugs in new drug discovery and development has been increasing. Seeking an effective research and development (R&D) strategy for FIC drug generation is a key issue for pharmaceutical company productivity. 

Conversely, FIC drug generation difficulties have been increasing. Novel promising drug targets are limited after long-term efforts of target-based drug discovery for creating FIC drugs [9], and 88% of proven drug targets are pursued by multiple pharmaceutical companies [10]. Many “druggable” molecular targets were already pursued, and “undruggable” targets remain for new drug discovery [11]. Therefore, pharmaceutical and biotechnology companies must urgently develop FIC drugs against these challenging molecular targets, possibly by utilizing modality technologies. The emergence of antibody medicines, and, more recently, nucleic acid medicines and gene therapies enable the approach against proteins without small molecule ligand pockets, mRNAs, and mutated gene replacement, respectively [12,13,14]. Developing next-generation antibody medicines, such as antibody–drug conjugates (ADC) and bispecific antibodies (BsAb), has provided a breakthrough in drug treatment [15,16]. Novel molecules, including medium-sized protein–protein interaction modulators and proteolysis-targeting chimeras, are examples of new modality technologies targeting “undruggable” proteins [17,18]. Accordingly, elucidating the effects of the evolution of modality technologies on recent FIC drug development is relevant. Thus, this study analyzed the chronological changes in the target families and modalities of FIC drugs approved by the FDA from January 2011 to December 2022.

Half of the scientifically innovative drugs approved by the FDA from 1998 to 2007 were discovered by small biotechnology companies, most of which are university start-ups [19]. The approach against challenging molecular targets has recently become more important in new drug development. However, who creates these innovative FIC drugs and the importance of the role of small biotechnology companies remain unknown. Thus, this study aimed to elucidate the fundamental mechanism that delivers novel technologies into new drugs by analyzing the origin of FIC drugs and highly productive small and medium enterprises (SMEs) that created more than one approved FIC drug.

## 2. Results

This study designated 168 drugs as FIC drugs among the 461 drugs approved by the FDA from 2011 to 2022, excluding imaging agents and new fixed-dose combinations of existing drugs. A slight upward trend overall was observed, although the number of approved FIC drugs varied from year to year from 2011 to 2022 (Figure 1). 

### 2.1. Target Families of FIC Drugs

The target family of FIC drugs approved by the FDA from 2011 to 2022 was analyzed in three periods, 2011–2014, 2015–2018, and 2019–2022 (Figure 2). In 2011–2014, the share of target families was 7.3% in receptor (G-protein coupled receptors (GPCR)), 7.3% in receptor (non-GPCR), 7.3% in the transporter, 7.3% in transmembrane protein (non-receptor), 34.1% in enzyme, 2.4% in transcription factor, 9.8% in secreted protein/peptide, 2.4% in mRNA, 0% in gene therapy, 7.3% in cell, 14.6% in non-human proteins and 0% in others. In 2015–2018, the share of target families was 5.6% in receptor (GPCR), 7.4% in receptor (non-GPCR), 1.9% in the transporter, 11.1% in transmembrane protein (non-receptor), 29.6% in enzyme, 0% in transcription factor, 16.7% in secreted protein/peptide, 5.6% in mRNA, 5.6% in gene therapy, 1.9% in cell, 9.3% in non-human proteins and 5.6% in others. In 2019–2022, the share of target families was 6% in receptor (GPCR), 7.5% in receptor (non-GPCR), 4.5% in the transporter, 7.5% in transmembrane protein (non-receptor), 17.9% in enzyme, 3% in transcription factor, 20.9% in secreted protein/peptide, 7.5% in mRNA, 9% in gene therapy, 1.5% in cell, 7.5% in non-human proteins and 7.5% in others. 

A previous study reported that the major target families of the FDA-approved drugs from the 1930s to 2013 were GPCRs, transporters, transcription factors, and enzymes [20]. This study reveals GPCRs, transporters, and transcription factors have a maximum share of less than 8%, 8%, and 3%, respectively, among all target families identified in these three periods. This indicates that these target families are no longer major FIC drugs target families in the last 12 years. Enzymes had a 34% share among all target families identified in 2011–2014, which substantially declined to 18% of the total in 2019–2022. Contrastingly, secreted proteins/peptides, mRNAs, gene deliveries, and other categories, including metabolites, sugar chains, and structural proteins, continuously increased from 10% to 21%, from 2% to 7%, from 0% to 9%, and from 0% to 7% in 2011–2014 to 2019–2022, respectively. These results indicate a shift in FIC drug targets from previous major target families to newer target families between 2011 and 2022.

### 2.2. Modalities of FIC Drugs

Non-major target families, considered “undruggable” because they lack the binding pocket for small molecules or are not proteins, are being increasingly used as targets for FIC drugs, possibly due to increased modalities that control targets that cannot be modulated using small molecules. Figure 3a summarizes the distribution of modalities used for FIC drugs during 2011–2014, 2015–2018, and 2019–2022. Small molecules remained predominant and comprised nearly two-thirds of the FIC drugs in 2011–2014. The percentage of antibody medicine more than doubled as seen in the previous period in 2015–2018, (from 16% to 37%). The antibody medicine modalities further diversified in 2019–2022 (Figure 3b). All antibody medicines approved in 2011–2014 and 2015–2018, except one ADC in each period, were conventional monoclonal antibodies. The approved FIC antibody medicines in 2019–2022 included two BsAb and two antibody fragments, which are next-generation antibody medicine modalities, in addition to four ADCs and nine conventional monoclonal antibodies (Figure 3b). This indicated diversified types of antibody medicine modalities compared to those in the previous periods. The ratio of nucleic acid medicine continuously increased with time and had a 7% share in 2019–2022 (Figure 3a). Gene therapies emerged in 2015–2018 and had an 8% share in 2019–2022 (Figure 3a). These results indicate diversified modalities used for FIC drugs have the increasing presence of next-generation antibody medicines, nucleic acid medicines, and gene therapies in 2019–2022 compared to 2011–2014, when small molecules remained predominant.

### 2.3. Originators of FIC Drugs

The above results indicate that the recent FIC drugs incorporate more challenging target molecules with various modalities, requiring new modality development. Many cutting-edge technologies are being developed in universities and technology start-ups. A previous study indicated the importance of small biotechnology companies, most of which were university start-ups, in creating innovative new drugs [19]. However, more recent data covering the drugs approved from 2011–2022 is lacking. This study revealed that more than half of the FDA-approved FIC drugs in 2011–2022 were created by SMEs (Figure 4a). The share of SMEs as original creators increased with time, with 62% of FIC drugs created by SMEs in 2019–2022 compared to 49% in 2011–2014. Among the total number of 98 SMEs that created FDA-approved FIC drugs in 2011–2022, 33 (34%) and 30 (31%) SMEs were established in the 2000s and 1990s, respectively, showing that relatively young companies based within the past 20–30 years have highly contributed to FIC drug generation (Figure 4b).

### 2.4. SMEs Creating More Than One FIC Drug in 2011–2022 and Their Drugs’ Modalities

Highly productive SMEs that created more than one FIC drug in 2011–2022 were identified to further investigate the contribution of SMEs to FIC drug generation (Table 1). Nine SMEs were identified. Agios Pharmaceuticals developed small molecule enzyme activators or inhibitors mitapivat, ivosidenib, and enasidenib. Alexion Pharmaceuticals developed recombinant proteins for orphan diseases sebelipase alfa and asfotase alfa. Alnylam Pharmaceuticals developed small interfering RNA (siRNA) medicines lumasiran, givosiran, and patisiran. BioMarin Pharmaceuticals developed peptide or recombinant proteins for orphan diseases vosoritide, cerliponase alfa, and elosulfase alfa. Bluebird Bio, Inc. developed gene therapy elivaldogene autotemcel, betibeglogene autotemcel, and idecabtagene vicleucel. Genmab is a company that was spun off from Medarex, which is a pioneer company of antibody medicines. It developed an ADC tisotumab vedotin and a monoclonal antibody medicine teprotumumab. Ionis Pharmaceuticals developed antisense oligonucleotide medicines nusinersen and mipomersen. PDL BioPharma developed monoclonal antibody medicines elotuzumab and mepolizumab. Sarepta Therapeutics developed antisense oligonucleotide medicines casimersen, golodirsen, and eteplirsen. Accordingly, all nine SMEs created drugs using the same single modality class. Six of the nine SMEs have technological expertise in recently emerging modalities, i.e., antibody medicine in Genmab and PDL, antisense oligonucleotide medicine in Ionis and Sarepta, siRNA medicine in Alnylam, and gene therapy in Bluebird.

These cases imply that companies with strengths in specific modalities have been highly productive in creating FIC drugs, supporting the importance of modality technologies in recent FIC drug development.

## 3. Discussion

FIC drugs have provided novel therapeutic options for physicians and patients. FIC drug development is becoming increasingly important due to increasing reimbursement pressure and a shift toward orphan diseases [1,2,5]. Contrarily, many “druggable” molecular targets have already been pursued, compounding the challenge of developing drugs against new targets [10,11]. This study reveals that the target families for FDA-approved FIC drugs in 2011–2022 have changed. The most significant classes of new drugs from the 1930s to 2013, including the GPCRs, transporters, and transcription factors [20], were no longer major, each with a maximum of only 3–8% share of all target families in the three analyzed periods. Enzymes, which is another major drug target family, still shared 34% of all target families in 2011–2014, but the share steeply dropped down to 18% in 2019–2022. Instead, secreted proteins/peptides, mRNAs, gene deliveries, and other categories, including metabolites, sugar chains, and structural proteins, have increased from 2011–2014 to 2019–2022, each with 21%, 7%, 9%, and 7% share in 2019–2022, respectively. These target families were previously considered “undruggable” due to the lack of small-molecule binding pockets and the infeasibility of gene delivery. These challenging drug targets are rapidly replacing the “druggable” target families as major FIC drug targets.

This study indicates the contribution of modality technology expansion to a shift in the approach toward challenging molecular targets. Antibody medicines can target molecules that are hard to be modulated by small molecules. Moreover, nucleic acid medicines can target mRNAs that were not previously considered drug targets. Gene therapies enable normal gene delivery and functional protein restoration in monogenic diseases. These three modalities have recently emerged. The share of antibody medicines in all target families increased from 16% to 37% from 2011–2014 to 2015–2018. Furthermore, the variety of antibody medicines increased in 2019–2022, including next-generation antibody medicines, such as ADC, BsAb, and antibody fragments, compared to conventional monoclonal antibodies. The shares of nucleic acid medicines, including antisense oligonucleotide and siRNA medicines, have continuously increased. Gene therapies emerged in 2015–2018 and had an 8% share in 2019–2022. This modality diversification enables “hard” drug targets to become more “druggable”, thereby creating new drugs within the challenging target families.

The changes in the target family are closely related to personalized medication advancement. One-fits-all types of drugs shifted to drugs with mechanisms tailored to the patient’s disease causes [3,4]. Hence, the demand to develop drugs targeting molecules that were difficult to regulate with conventional drugs is growing. This could have caused the increased share of previously “undruggable” targets such as secreted protein/peptide and mRNA.

The above results indicate that pharmaceutical and biotechnology companies that develop innovative new drugs should increase their investment in developing new modalities. The competitiveness of drug discovery used to stem from biology. The academic discovery of new physiological molecules and biological functions highly contributed to new drug development. This study indicates that the competitiveness of innovative new drug development has shifted to modality technologies. All nine SMEs that created more than one FIC drug in 2011–2022 focused on a single modality in their approved drugs. New modalities help modulate novel drug target molecules, which otherwise could not be conducted by existing modalities. Access to these new modality technologies might increase the chances of creating FIC drugs. Six of the nine SMEs demonstrating high performance in creating the FIC drugs approved in 2011–2022 are developed by companies with strong capabilities in recently emerging modalities, such as antibody medicines, nucleic acid medicines, and gene therapies.

Many drug technologies represented by new modality technologies start from academic research and transfer to start-ups. This study revealed that SMEs contribute to creating over half of the FIC drugs, indicating the continuous importance of SMEs in innovative drug discovery. Relatively young SMEs established in the 1990s and 2000s created approximately two-thirds of the FIC drugs created by SMEs and approved in 2011–2022, indicating that biotechnology start-ups are being actively established and are translating scientific discoveries into innovative drug development. Academic research should increasingly focus on developing modality technology from the perspective of innovative new drug discovery. Young biotechnology start-ups can be strong drivers to bring new technologies into innovative drug development. Therefore, translational mechanisms to apply new modality technologies into new drug development, such as financial and regulatory support for establishing the modality technology start-ups, should also be strengthened. Drug developers should increase their access to academic discovery of novel modalities and incorporate them into their R&D to enhance FIC drug development. Biotechnology start-ups in the United States and Europe with innovative modalities receive significant long-term investments from venture capital firms and large pharmaceutical companies, supporting their R&D [21]. The development of modalities in start-ups may be promoted by continuing this mechanism. Conversely, such a start-up ecosystem has not yet fully developed in Japan [22], and the government should take strong leadership in promoting start-up initiatives and facilitating the matching between academia and existing pharmaceutical companies in terms of modality technologies to translate academic modality technologies into drug discovery.

This study is believed to fill the gap in existing research and provide readers interested in innovative new drug development with new insights and perspectives. This study aims to investigate the changes in technology used for FIC drugs and indicate directions for efficient FIC drug development in the future. Reviews introducing newly FDA-approved drugs have been published, reporting on disease area and modality proportions [23,24]. However, the long-term technological trends in FIC drugs had little comprehensive assessment. Reviews introducing trends in new drugs have been published for specific disease areas [14,25], but the research that comprehensively investigates FIC drugs across diseases and quantitatively reveals the technological trends of FIC drugs as a whole is lacking. Many papers are reporting on the progress of modalities used in new drugs, but most of them focus on specific modalities, e.g., [26,27], and few papers report on the overall changes in modality technology. Examining the types of companies contributing to FIC drug creation is an important perspective to identify drivers of innovative drug development and suggest directions for the efficient development of FIC drugs in the future. However, similar reports are limited to specific disease areas or old [19,28], and comprehensive data on recent trends have not been reported.

This study has several limitations. First, it does not capture trends of drugs approved outside the United States because it examines FDA-approved drugs. The drug discovery innovation systems vary by country [22], and this study does not fully consider the differences between countries. Second, it does not examine the trends in modality technologies and drug targets currently under development due to the focus on approved drugs. The process from research to drug approval takes 10–15 years, thus this study does not adequately address the more recent technological trends. Third, it cannot follow the evolution of best-in-class drug development because it focuses on FIC drugs. Best-in-class drugs are known to provide unprecedented value to patients [29], and information about their technological trends and originators is crucial in suggesting future directions for drug development. However, this study does not cover that aspect.

## 4. Materials and Methods

### 4.1. Definition of FIC in This Study

The definition and identification methods of FIC drugs vary depending on the articles [1,5,7,19]. Here, drugs are designated as FIC based on their direct molecular target. The drug that modulates a molecular target not previously modulated by other already approved drugs during its FDA approval process, is defined as an FIC drug. If two drugs modulate the same biological pathway but act on different molecules, they are designated as drugs belonging to different classes. If two drugs act on the same molecule with different binding sites/modes, they are considered drugs belonging to the same class, and the one that was approved first was defined as an FIC drug. If the drug targets more than one molecule, the drug was defined as an FIC drug only if the targets included molecule(s) that had not been targeted by previously approved drug(s).

### 4.2. Identification Method of FIC Drugs, Their Target Family, Modality, and Origin

New FDA-approved drugs from 2011 to 2022, excluding imaging agents and new fixed-dose combinations of existing drugs, were identified from New Drugs at the FDA (https://www.fda.gov/drugs/development-approval-process-drugs/new-drugs-fda-cders-new-molecular-entities-and-new-therapeutic-biological-products accessed on 1 March 2023). Gene and cell therapies approved by the FDA from 2011 to 2022 were identified from Approved Cellular and Gene Therapy Products (https://www.fda.gov/vaccines-blood-biologics/cellular-gene-therapy-products/approved-cellular-and-gene-therapy-products accessed on 1 March 2023). The molecular target of each drug was searched in DrugBank Online (https://go.drugbank.com/ accessed on 1 March 2023). DrugBank Online has a unique function that can search for other drugs that target the same molecule. The drug was identified as FIC if no other drugs target the same molecule and were previously approved. DrugBank Online also lists drugs that target the same molecule indirectly or old drugs that are considered to act on many molecules broadly in some cases. Those drugs were not considered preceding drugs. Human cord blood hematopoietic progenitor cell products were created by many companies and academic institutions in cell therapy, and the product approved the earliest among them was identified as FIC. Chimeric antigen receptor (CAR)-T cell therapies were categorized as gene therapy (not cell therapy). There were two CD19-directed CAR-T and two B-cell maturation antigen (BCMA)-directed CAR-T cell therapies in gene therapy. The product approved earlier in each mechanism was identified as FIC. The drug was excluded from the analysis if a direct molecular target of the drug was not identified.

The target family of the molecular targets of each drug was searched on the web and in scientific papers. The target family of the drug was categorized as gene delivery if the drug is a gene therapy. The target family of the drug was categorized as a cell if the drug is a cell therapy. The drug modality of each drug was identified in DrugBank Online. The origin of each drug was identified on the “Asu no Shinyaku (Tomorrow’s New Drug)” website (https://technomics-info.com/jsp/top.jsp accessed on 1 March 2023), which is a database that provides originator information of approved and clinically developed drugs. “Asu no Shinyaku” is the database that records drug discovery and development information collected from press releases, news reports, scientific meetings, and scholarly papers. Each record is sorted by the drug name and includes the name of the organization conducting the drug research and the name of the company conducting the clinical development. Companies originate drugs and then license them to different companies in some cases. Because the “Asu no Shinyaku” database separately identifies the company or academic institution that originally created the drug and the company that in-licensed, developed clinically, and obtained approval if it was the case, this study was able to identify the originator of the drug, as distinct from the company that developed it. The originator information was double-checked by the Internet search to increase the accuracy. Furthermore, the top 50 pharmaceutical companies that demonstrated the highest revenue in 2021 were identified on the Drug Discovery & Development website (https://www.drugdiscoverytrends.com/pharma-50-the-50-largest-pharmaceutical-companies-in-the-world-for-2022/ accessed on 1 March 2023). The company was classified as a large enterprise (LE) if an originator company was included in the top 50 companies; otherwise, the company was classified as SME. The top 50 pharmaceutical companies in the year before the acquisition were checked from the website which provided the 2012–2019 yearly sales ranking of pharmaceutical companies (https://www.rankingthebrands.com/The-Brand-Rankings.aspx?rankingID=370&year=1270 accessed on 1 March 2023) if an originator company does not exist in 2021 (e.g., AveXis was acquired by Novartis in 2018, Shire was acquired by Takeda in 2019), and the company was classified as LE if the originator company was in the top 50 before the acquisition, and classified as SME if not. The location and establishment year of the originator companies was identified in the CB Insights (https://www.cbinsights.com/ accessed on 1 March 2023), the company’s homepage, and web information.

## 5. Conclusions

The target family of FIC drugs has shifted to molecules previously considered challenging as drug targets and modality diversification facilitates drug discovery toward the challenging molecules. Investment in modality technologies and facilitating mechanisms to translate academic modality technology development to start-ups might be important for enhancing FIC drug development.

## Figures and Tables

**Figure 1 pharmaceutics-15-01794-f001:**
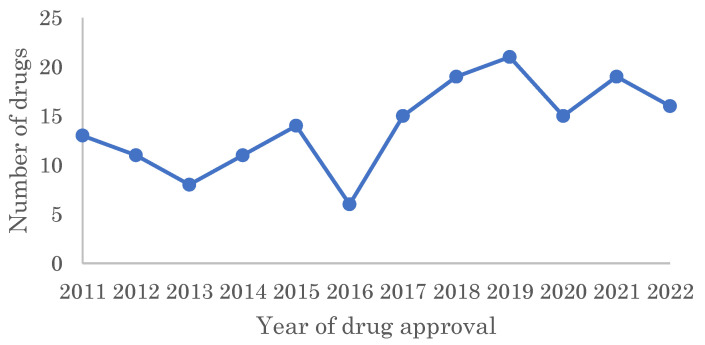
Chronological analysis of the number of FIC drugs approved by the FDA from 2011–2022.

**Figure 2 pharmaceutics-15-01794-f002:**
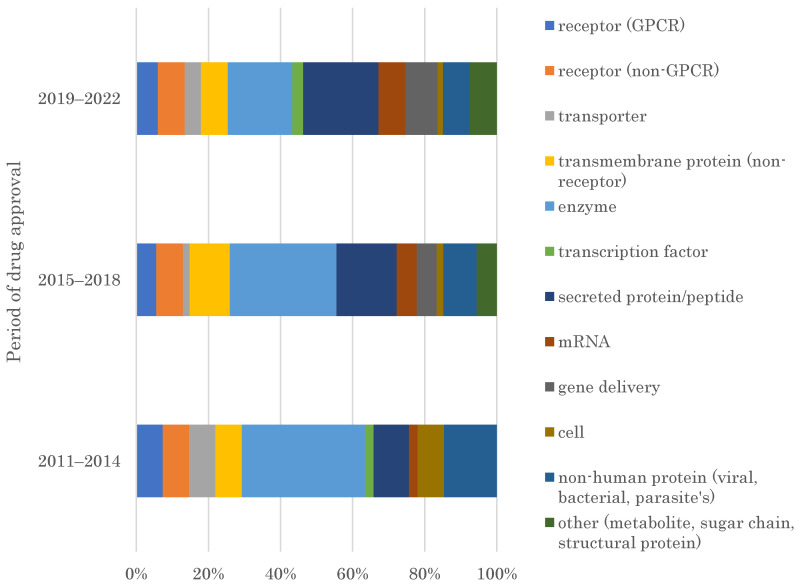
Chronological analysis of the distribution of target families of FIC drugs approved by the FDA from 2011–2022 in the three periods, 2011–2014, 2015–2018, and 2019–2022.

**Figure 3 pharmaceutics-15-01794-f003:**
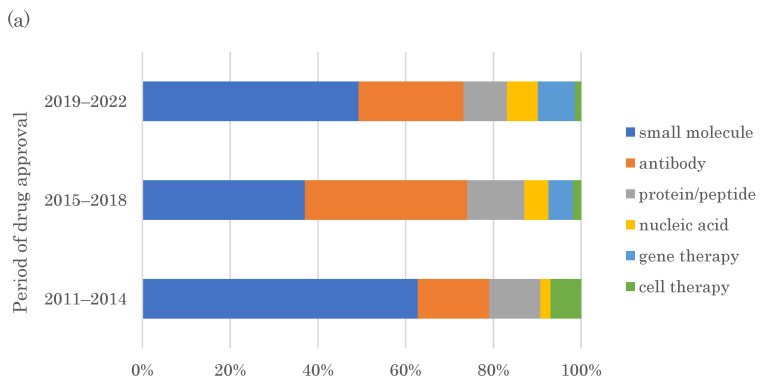
Chronological analysis of the modalities of FIC drugs approved by the FDA from 2011–2022 in the three periods, 2011–2014, 2015–2018, and 2019–2022. (**a**) Distribution of the FIC drug modalities. (**b**) Types of antibody medicine modalities.

**Figure 4 pharmaceutics-15-01794-f004:**
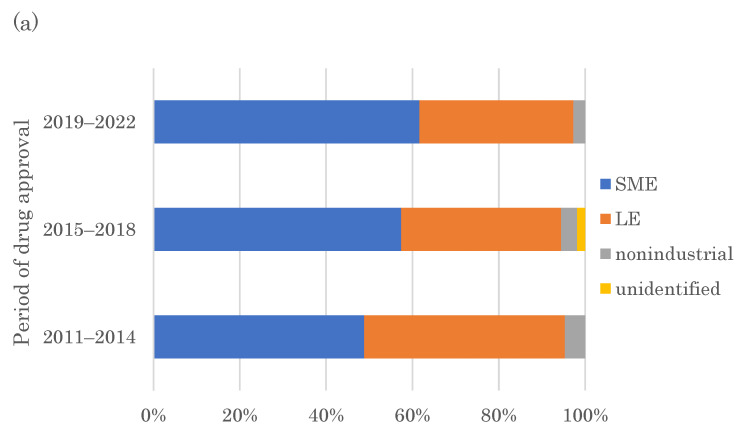
Analysis of the originators of FIC drugs approved by the FDA from 2011–2022. (**a**) Chronological analysis of the ratio of SMEs and LEs in the three periods, 2011–2014, 2015–2018, and 2019–2022. (**b**) Distribution of the establishment periods of the SMEs that originally created FIC drugs.

**Table 1 pharmaceutics-15-01794-t001:** Highly productive SMEs that created more than one FIC drug approved by the FDA from 2011–2022.

Company	Drug Name	Modality
Agios Pharmaceuticals	mitapivat	small molecule
	ivosidenib	small molecule
	enasidenib	small molecule
Alexion Pharmaceuticals	sebelipase alfa	recombinant protein
	asfotase alfa	recombinant protein
Alnylam Pharmaceuticals	lumasiran	siRNA
	givosiran	siRNA
	patisiran	siRNA
BioMarin Pharmaceutical	vosoritide	peptide
	cerliponase alfa	recombinant protein
	elosulfase alfa	recombinant protein
Bluebird Bio, Inc.	elivaldogene autotemcel	gene therapy
	betibeglogene autotemcel	gene therapy
	idecabtagene vicleucel	gene therapy
Genmab	tisotumab vedotin	antibody–drug conjugate
	teprotumumab	monoclonal antibody
	daratumumab	monoclonal antibody
Ionis Pharmaceuticals	nusinersen	antisense oligonucleotide
	mipomersen	antisense oligonucleotide
PDL BioPharma	elotuzumab	monoclonal antibody
	mepolizumab	monoclonal antibody
Sarepta Therapeutics	casimersen	antisense oligonucleotide
	golodirsen	antisense oligonucleotide
	eteplirsen	antisense oligonucleotide

## Data Availability

The data that support the findings of this study are available from the corresponding author upon reasonable request.

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
