# Peer review of "Chronological Analysis of First-in-Class Drugs Approved from 2011 to 2022: Their Technological Trend and Origin"

_pharmaceutics, 2023, doi:10.3390/pharmaceutics15071794_

Round 1
Reviewer 1 Report
See attachment for my feedback

Reviewer 2 Report
The authors introduced the chronological analysis of first-in-class drugs approved from 2011–2022. It seems to be a review paper. Comments:
1. The sub-titles of part 2~4 should be revised to summarize the content of each part.
2. Authors should clearly indicate what new knowledge is gained from this work in contrast to the previous review papers published in this area. Authors are encouraged to cite the missing review papers related to the topic.
3. Lack of description and explanation for results presented in the tables. The authors have only briefly transcribed several descriptions from the original references. A lack of systematic description/explanation has made it very difficult to fully understand a figure showing multiple reaction steps/components. The names of drugs should be provided.
4. The authors should have offered some perspectives on the selected topic including possible barriers.
5. There are clearly some language issues, giving rise to ambiguous statements and unclear expressions.
Minor editing of English language required
Reviewer 3 Report
In this manuscript of " Chronological analysis of first-in-class drugs approved from 2011–2022: Their technological trend and origin", it fully demonstrates the authors' accumulation of knowledge in the field and suggests new trends for other scholars to follow. But this needs revision and the author reply comments properly before accepted. The comments and questions are as follows:
1. What is the impact of the amount of sulfate and molecular weight of fucoidan on its activities?
2. What were the previous major target families for FIC drugs, and how have they changed in recent years? What types of molecules have seen an increased share as target families for FIC drugs?
3. How has the share of nucleic acid medicine changed over time? What does the increase in nucleic acid medicine share suggest about FIC drug development?
4. Based on the findings presented, what recommendations would you make to enhance the development of FIC drugs and support the growth of SMEs in this field? How can facilitating mechanisms for translating academic modality technologies to start-ups support FIC drug development?
5. How has the shift in target families of FIC drugs impacted the development of personalized medication? What factors have contributed to the increased share of secreted proteins/peptides and mRNAs as target families for FIC drugs?
Above all, I suggest this can be accepted before making revision or answer the questions properly.
Round 2
Reviewer 2 Report
Accept in present form.
Reviewer 3 Report
Accepted